# Applying Religious Studies Discourse during Wartime: On Katō Totsudō's Discussion of Religious War

Akira Nishimura 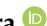

Graduate School of Humanities and Sociology, The University of Tokyo, Tokyo 113-0033, Japan; aquillax@gmail.com

**Abstract:** This paper argues how the study of religion in Japan influenced the nation's conduct during the Asia-Pacific War (1931–1945). Specifically, the paper addresses the wartime national indoctrination texts of Katō Totsudō 加藤咄堂 (a.k.a. Katō Yūichirō 加藤熊一郎, 1870–1949). Although Katō was not strictly a religious scholar, analyzing Katō's texts is significant in understanding the influence of religious theory and religious studies discourse on Japanese society during the war. To illustrate this point, this paper introduces previous studies that have discussed the movements of religious scholars during the war. It then clarifies the significance of discussing Katō's texts, followed by an introduction to what has been revealed about Katō so far. The paper then examines Katō's wartime texts that discuss the relationship between war, faith and the readiness to die. The East–West comparison of views of life and death used by Katō was characterized as a wartime application of comparative religion. It was intended to emphasize Japan's superiority over other countries. Such features agitated Japanese readers to proactively enter the fight to the death through spiritual mobilization in a total war system.

**Keywords:** Katō Totsudō 加藤咄堂; religious studies; Japanese religion; Buddhism; Asia–Pacific War; view of life and death; preparing to die

## 1. Introduction

This paper argues how the study of religion in Japan influenced the nation's conduct during the Asia-Pacific War (1931–1945). In *Savage Systems*, David Chidester repositions the comparative religion of the 19th century into the broader history of the relationship between European imperialism and colonial areas, pointing out that the knowledge of various religions gained from contact situations at the frontiers during colonization was systematized in the "imperial comparative religion" and that this academic knowledge was then reimported to the former frontiers in the 20th century and applied to racial segregation policies as the "apartheid comparative religion" (Chidester 1996). Although the scope of this paper does not have such a historical range, the question of how academic knowledge affects a violent situation cannot be avoided in a critical examination of the history of scholarship.

The study of religion in Japan began around 1900. For example, Japan's first department of religious studies was established by Anesaki Masaharu 姉崎正治 (1873–1949) at Tokyo Imperial University 東京帝国大学 in 1905. The Japanese Association for Religious Studies 日本宗教学会 was established in 1930, on the eve of Japan's entry into the Asia-Pacific War, also known as the Fifteen Years war. The interest of this paper is the impact of such religious studies on the war in the same period. Specifically, I will focus on the wartime national indoctrination texts of Katō Totsudō 加藤咄堂 (real name: Katō Yūichirō 加藤熊一郎, 1870–1949).

Katō was a writer and intellectual who taught cultivating the mind in an easy-to-understand manner to the public. He was appointed as the chief editor of the Buddhist

newspapers *Meikyō shinshi* 明教新誌 and *Chūgai nippō* 中外日報. Additionally, he participated in the New Buddhism Movement 新仏教, which criticized the existing Buddhist sects and advocated "free inquiry". However, he was not a Buddhist priest but a layman who explained the significance of Eastern and Buddhist thoughts and Zen training methods in plain language from his perspective. Another characteristic of his narrative is his incorporation of the rhetoric of Western studies and modern thought, in which there was room for religious studies knowledge to be utilized. He published more than 200 books and gave more than 200 lectures a year at the peak of his career. From 1924 onward, he was a central figure in the Central Federation of Edification Institution 中央教化団体連合会 and was actively involved in national indoctrination after the Great Kantō Earthquake and during the wartime period (Okada 2016, pp. 105–6; Ma'ekawa 2015, p. 420). Katō was a contemporary of Anesaki and had contact with him and other religious studies scholars in various places. For example, inspired by the Parliament of the World's Religions held in Chicago in 1893, a conference of religious leaders, 宗教家懇談会 was held in Japan in 1896. Katō attended the meeting, and among the 48 participants were Anesaki and Kishimoto Nobuta 岸本能武太 (Ma'ekawa 2015, pp. 79–80). At that meeting, Anesaki and Kishimoto discussed the establishment of a Society for Comparative Religion (Isomae and Fukasawa 2002, p. 246). Katō was also a significant member, along with Anesaki, of the Kiitsu Kyōkai 帰一協会, which was organized in 1912 under the motto "Unity of Class, Nation, Race, and Religion", and the Japan Religious Dialogue 日本宗教懇話会, which was formed in 1924 and later became the Japan Religious Society 日本宗教協会.

He was hardly a scholar of the discipline, despite these exchanges with religious scholars. In his voluminous writings, however, Katō occasionally introduces theories of religious studies as well as philosophy and other Western thought[1]. He can be regarded as a person who spoke publicly about religious studies during the period from the dawn of the discipline in Japan to the mid-war period. However, that fact has not been actively pointed out in previous studies. Therefore, an examination of Katō's texts is essential in capturing the influence of religious theories and religious studies discourse on Japanese society during the war. In the following sections, I introduce previous studies that have discussed the words and actions of religious scholars during the war, clarify the significance of referring to Katō's texts (Section 2), and then introduce findings about Katō from previous research (Section 3)[2]. Then, I will specifically examine *Sensō to shinkō* 戦争と信仰 (hereafter *War and Faith*), published in 1938, and *Shi ni chokumen shite* 死に直面して (hereafter *Confronting death*), published in 1944 (Sections 4 and 5) (Katō 1938, 1944).

## 2. Background of the Research Question

Suzuki Norihisa 鈴木範久, the author of the pioneering study on the history of Japanese religious studies (Suzuki 1979), discussed the social comments by religious studies scholars in the journals during the Pacific War in the early 1940s (Suzuki 2005). Specifically, Suzuki traced the statements by leading scholars of religion at the time in religious review journals such as *Shūkyō kōron* 宗教公論 (Public Opinion of Religion) and *Shūkyō kenkyū* 宗教研究, the official journal of the Japan Society for the Study of Religion. He pointed out that many of their statements were characterized by the introduction and commentary on the current situation of religion in the Southward 南方 (i.e., Southeast Asia, South Asia, and the Pacific Islands), which the Japanese military occupied during the war. Additionally, while they emphasized academic research and the understanding of local religions, their research strengthened the character of pandering to the Japanese government.

Suzuki omitted the statements of some scholars, such as Hamada Honyū 浜田本悠 (1891–1971), the editor of *Shūkyō kōron* and one of the students of Anesaki, because he considered them too firmly integrated with the current state of the war affairs.

In contrast, Nishimura Akira 西村明, using articles in *Shūkyō kenkyū* as his primary source material, summarized the situation of the Japanese Association of Religious Studies during the war period and research about the war that appeared in that journal (Nishimura 2008). In 1938, the year following the outbreak of the Sino-Japanese War, the academic

conferences of the Japanese Association for Religious Studies included references to the war situation in the opening address and "national rites", such as prayers for victory over friendly forces and commemoration of the war dead in the conference program. This indicated the expected contribution of scholarship to the revitalization of the national spirit and psychological governance in the colonies and occupied territories. Nishimura pointed to the "strange congruence" between such political expectations and the academic task of understanding various religions. A few years after the appearance of such statements and rituals, Nishizawa Raiō 西澤頼応 (date of birth and death unknown) and Hamada Honyū submitted papers on the war in the journal. They discussed the religious beliefs and rituals on the battlefront and the home front in the context of religious history. Further, their study presented the cultivation of religious faiths as a pressing issue in the conduct of war.

These studies have significance as critical examinations of religious discourses during wartime that have been largely neglected in the past. However, they are limited in terms of their social impact. The materials discussed here are intended for religious scholars and religious figures. While they can be used as a reference for the impact of the war on policy, it is difficult to grasp their effect on and relationship with society during the war. Therefore, the next question that remains to be examined is the extent to which the discourse of these religious researchers reached the military and the public.

Therefore, I will approach this issue from the overlap between books related to religious studies that may have been referred to in military schools and the list of publications ordered confiscated by the General Headquarters of the Supreme Commander for the Allied Powers (GHQ/SCAP) after the war. Of course, it is also necessary to examine reactions from the recipient side but, due to material limitations, this point will not be addressed in this paper. It will be left as an issue to explore in the future.

First, as books related to the military academy, the "Books of the Former Naval Academy 旧海軍大学校図書" in the collection of the Japan Coast Guard (JCG) Academy are among those whose whereabouts can be confirmed at present. After Japan accepted the Potsdam Declaration on 14 August 1945 and lost the war, the Japanese Army and Navy were disarmed by GHQ/SCAP on 30 November 1945. In December 1951, about 8000 books from the former collection of the Naval Academy, an educational institution for senior officers of the Navy, established in 1888, were stored in the Rare Book Collection of the JCG Academy Library. In 2004, it was organized to be viewable.

The following is a rough classification of books on religion from the "List of Books of the Former Naval Academy: Critiques on policy toward Asia and the current situation by Ōtani Kōzui 大谷光瑞 (1876–1948)[3], works of literature that discussed the Japanese spirit 日本精神, the Imperial Way spirit 皇道精神, and Shinto thought[4], and those that discussed Asian thought and peoples by Takakusu Junjirō 高楠順次郎 (1866–1945) and Hori Ichirō 堀一郎 (1910–1974). Notably absent from these categories are two books by Katō Totsudō that deal directly with the subject of war from a religious perspective or with topics related to war. One is *War and Faith* (Katō 1938) and the other is *Confronting death* (Katō 1944). We can find three volumes of Katō's writings in the collection of the library with the added *Nihon shisō to daijō seishin* 日本思想と大乗精神 (*Japanese spirit and Mahayana mind*) (Katō 1934), which discusses the relationship between the Japanese spirit and Mahayana Buddhist thought.

All three books are also included on the list of publications confiscated during the GHQ occupation. The order to confiscate was issued in a memorandum from the Commander-in-Chief of the Allied Forces dated March 11, 1946, and 46 additional memoranda were issued, totaling over 7700 prewar and wartime publications ordered to be confiscated. The Japanese government, under the jurisdiction of the Cultural Affairs Division of the Social Education Bureau of the Ministry of Education 文部省社会教育局文化課, targeted books in bookstores, warehouses, and other distribution channels through each prefectural board of education, excluding books owned by individuals and libraries (Monbushō Shaka'ikyōikukyoku 1949).

This catalog also included many of the works in the previous list. In addition to the three books listed above, Katō's works include *Hataraku kokoro* 働くこころ (*Working mind*) and seven volumes of *Nihon seishin bunken sōsho* 日本精神文献叢書 (*Japanese spirit literature collection*), compiled by Katō (Katō 1942; Katō 1938–1940).

The works of religious scholars mentioned in Suzuki and Nishimura's previously mentioned studies that appear on this confiscated list are as follows: Hori Ichirō, *Indo minzokuron* 印度民族論 (*India's Ethnicity*), Furuno Kiyoto 古野清人 ed., *Nanpō mondai jukkō* 南方問題十講 (*Ten lectures on the issues of the Southward occupied areas*), Hiyane Anjō 比屋根 安定, *Nihon shukyō zenshi* 日本宗教全史 (*Comprehensive Japanese history of religion*) Volume1 (Ancient era). Except for the historical narrative of Hiyane at the end, which discussed the continuity between myth and ancient history, the works of Hori and Furuno correspond to commentaries and policy recommendations on the religious situation in the Southward, as summarized by Suzuki (Hori 1940; Furuno 1942; Hiyane 1941).

In contrast, Katō's writings are characterized by their focus on national indoctrination, centering on Buddhist thought with knowledge of religion in general. Katō is not a scholar or specialist in religious studies. However, he is a remarkable figure regarding how the knowledge of religious studies was shared socially in Japan in the first half of the 20th century. It is also significant to discuss Katō's discourse regarding how understanding and attitudes toward war were propagated based on such scholarly knowledge.

### 3. Previous Research on Katō Totsudō

Katō's role in the social sharing of academic knowledge has already been pointed out in another discussion. In his commentary on *Minkanshinkōron senshū* 民間信仰論選 集 (*Selection of folk faith studies*), Nishimura Akira argued that since the late 19th century, when the concept of "folk faith" was coined by the founder of Japanese religious studies, Anesaki Masaharu, there has been a growing body of research in related fields in the first half of the 20th century. The book traces how the concept was applied to each study and social context in analysis and religious discourse (Nishimura 2016). In this context, I will focus on Katō's *Minkanshinkō-shi* 民間信仰史 (*History of folk faith*) (Katō 1925), which, after Anesaki's discussion, presented the concept of "folk faith" and then summarized it through a broader search of sources. Furthermore, in a novel way, this book regarded folk beliefs as traditional ethnic cultures in place of the Enlightenment view of them as superstitions, as seen at the end of the 19th century.

As mentioned earlier, due to not being a professional religious researcher, there has not been much discussion about Katō in previous studies. However, the few references and issues raised in previous studies are essential to the discussion in this paper and will be introduced below. Ishii Kōsei 石井公成 cited Katō's *War and Faith* as an example of a book that advocated war and was widely read but is not available in university libraries (Ishii 2020, p. 5). Citing data from Cinii Books, Ishii pointed out that the book is held only at the JCG Academy Library, two other university libraries, and two public libraries. He then stated, "the main reason for this situation is that the Ministry of Education issued a directive immediately after the defeat of the war to destroy blatantly militaristic literature," and insisted on the need to clarify how Buddhism was swallowed up and expanded by Japanism 日本主義.

Ma'ekawa Michiko 前川理子 questioned the relationship that religious scholars, not necessarily limited to those who were members of university academia, and their ideas had with the state and society (Ma'ekawa 2015). Her work discussed Katō in a fragmented manner. In Section 3, which examined the academic system of religious studies established by Anesaki Masaharu, Katō was mentioned as a participant in the Religious Leaders' Roundtables 宗教家懇談会 in 1896 and 1924 and in a lecture on the Cultivation of Civil Power 民力涵養講演会, held after 1919, following the rice riots 米騒動. Section 5 discussed the review of deliberations in various advisory committees established in the Ministry of Education since the Taishō Era; Ma'ekawa mentioned Katō as one of the religious scholars and comradely religious people involved in these bodies. As a director of the Central

Federation of Edification Institutions 中央教化団体連合会, Katō was introduced as "a widely recognized activist for the personality building based on a Buddhist background, with many works to his credit" (Ma'ekawa 2015, p. 386).

Shimazono Susumu 島薗進, in his essay on the concept of Death and Life Studies, discussed Katō's discourse as the first appearance of the term "view of life and death" in modern books (Shimazono 2008, p. 14). Shimazono described the great response to Katō's *Shise'ikan* 死生観 (*Views on Life and Death*), published in 1904, which led to the publication of an enlarged edition the following year, besides *Daishiseikan* 大死生観 (*Great Views on Life and Death*), in 1908. The term "view of life and death" has been recalled periodically since then, and he stated that "during the Asian-Pacific War, several books on life and death were published to encourage young men to go on the warpath" (Shimazono 2008, p. 14).

In contrast to these fragmentary references, Jolyon Thomas provided a coherent discussion of Katō (Thomas 2014). He focused on developments after the dissolution of the Shin Bukkyōto Dōshikai 新仏教徒同志会 (Fraternity of New Buddhists) and the prestige and political influence of former members as they entered middle age in the late Taishō and early Shōwa periods (approximately from the late 1910s to the 1930s). Katō was a prime example of this rise to influence.

Katō was a lay Buddhist who was not ordained as a priest in any Buddhist sect. However, he gained the acquaintance of the Ministry of Home Affairs bureaucrats, who oversaw religious administration. He was also used as a representative intellectual of the Buddhist circle in Japan for his social teaching activities. With his natural erudition, literary talents, and gift for storytelling, he engaged in educational activities based on Buddhist thought to mobilize the spirit for nation-building and eradicate superstition.

Thomas's primary interest lay in the modern organization and distribution of "religion", which emerged from the political and religious relations, the rivalry between old and new Buddhists and the criticism of superstition from an Enlightenment perspective (Thomas 2014). Thus, the scope of his argument did not include Katō's works during the Asia–Pacific War period, the focus of this paper. As Ishii pointed out in the first half of this chapter, from the perspective of looking at the involvement of Buddhism and other religions in the conduct of war, it is necessary to focus on Katō's words and actions around those days. As Shimazono mentioned, it is also noteworthy that there is a group of writings that expounded on the views of life and death to inspire young men to go to war.

Thus, this paper will examine Katō's arguments regarding the relationship between war and religion, focusing on his two works, *War and Faith* and *Confronting death*, while also referring to other works published during the war period.

## 4. *War and Faith*

*War and Faith*, published in 1938, consists of five chapters and two appendices. In the preface, Katō stated that the relationship between faith and war should be considered a matter of interest. He examined it from a historical point of view to "contribute to the reflection of the modern mind" (Katō 1938, p. i).

In the introduction, the book's interest is expressed more specifically. Namely, Katō stated that war as an expression of power and religion as a manifestation of love seems to have nothing to do with each other. However, never since ancient times has the religious mind been more active than in times of war. Katō asked, "How is war, which kills people, related in the first place to religion, which saves people?" (Katō 1938, p. 2). He then described how people prepared to fight with determination through mental discipline and prayers to the deities and Buddhas, as reported under the circumstances of the Sino–Japanese War. He compared this attitude to the faith of the samurai of pre-modern times. This emphasis on decisive resolve is the common theme in *Confronting death*, discussed later.

### 4.1. "Immortal Belief Even after Death"

In the section "Immortal Belief even after Death" in Section 1, "Preface", Katō argued that in war, we stand on the border between life and death, and we cannot fight if we are averse to end. Therefore, a "spirit that transcends life and death" is the basis for fighting bravely. He described this spirit as a "spirit of selflessness" and grounds it in religious belief. Katō's concept of "religion" here refers to preaching a detailed morality of life and death and an attempt to make people realize or believe in it. He cites the Japanese spirit of "death but not dying" as the most precious and glorious belief that emerges at the boundary of life and death in actual combat. He argues that this is recognized in the specific situation of the military, who died calmly by yelling "banzai" and facing the Japanese imperial palace in the East from the battlefield on the Chinese mainland (Katō 1938, p. 4).

Although the expression "even in death, never die" seems to be a contradiction in terms, it is explained in more detail as a belief that is not interrupted by the death of an individual life but is succeeded in continuity from ancestor to descendant. The inner substance of this belief is the view of nation protection and the idea of justice.

The former refers to a "kind of religious nationalistic belief" that the heroic spirits of the fallen will protect the country (Katō 1938, p. 6). A typical precedent of this belief is the "Shichisei Hōkoku no Shisō 七生報国の思想" of the brothers Kusunoki Masashige and Masasue 楠木正成・正季. They were loyal to Emperor Godaigo 後醍醐天皇 of the Southern Dynasty and died in a battle between the Northern and Southern Dynasties in the 14th century. Masashige swore that he would be "reborn (changed) into a human being seven times and destroy" the Northern Court as "national pirates." Katō argues that this mind of "dying but never die" was passed on to thinkers of the Son'nō Jō'i Undō 尊王攘夷運動, the movement advocating reverence for the Emperor, and the expulsion of foreigners in the late Edo period, such as Yoshida Shōin 吉田松陰 (1830–1859) and Fujita Tōko 藤田東湖 (1806–1855) (Katō 1938, p. 5).

Regarding the latter's sense of justice, based on the mythical historical view that Japan, unlike other countries of the same era, has a history of nearly 3000 years of the regime, and in light of its history of overcoming critical national difficulties and sustained development, he explains that the ongoing conduct of war is "unavoidable for the sake of eternal peace in the Orient" and just war that "takes up arms and dispels the demons for the welfare of the world's humanity." He adds that the sense of justice is the belief that we should behave in a manner unashamed of heaven and earth (Katō 1938, p. 8).

With this religious worldview as a premise, the chapters that follow address "Beliefs of a Divine Japan" (Section 2), "The Buddhist View of War" (Section 3), "Samurai and Faith" (Section 4), and "Emerging Mind and Faith" (Section 5).

### 4.2. "Belief in a Divine Japan"

Section 2 discusses the belief that the gods will protect their friendly forces and country and always win the war because Japan is a divine nation. After tracing the description of Japan as a "divine nation" in ancient history books, Katō cites the mythological story of the founding of the country, the "unbroken Imperial line", and the existence of the three sacred imperial treasures as the basis for Japan's status of a holy nation. In addition, he mentions the presence of the emperor, who is revered as a living god, and the unique political form of unanimity in rituals and politics.

In the same chapter, the section "Divine nation and War" introduces the idea that weapons have been kept at shrines as sacred treasures and the ancient respect for oracles when going to war. However, the section "Aspirations of the Divine nation" asserts that "the Divine nation never likes to fight" (Katō 1938, p. 19). Still, Japan's national inevitability is to advance externally toward the realization of a united world ideal state such as "Eight corners of the world under one roof 八紘一宇" as enshrined in the ideology of the construction of the Great East Asia Co-prosperity Sphere 大東亜共栄圏 at that time.

Focusing on the nature of the divine nation, the political and religious history of introducing Confucianism and Buddhism in ancient times, the Mongol invasion in the

Middle Ages, and the Christian mission and proselytizing in the early modern period are described. It then develops the argument that the idea of a holy nation is inclusive. In that context, Christianity was once banned because of its exclusiveness and the Spanish invasions behind its propagation after the Meiji Restoration. However, the ban was gradually lifted, and the Christian side took the attitude that worship of Japanese deities was a profession of patriotism and consistent with the doctrine of the only God. Katō saw this Christian alteration as a "sign of accommodation to the Divine nation" (Katō 1938, p. 55).

In the same chapter, Katō also mentions divine punishment as an idea of a holy nation associated with the conduct of war. He asserted that the belief that gods would protect friendly forces, said at the beginning of Section 2, is based on the idea that "our army by the divine will" and that oracles and divination were used in ancient times to inquire about this (Katō 1938, p. 56). With the development of moral values, it was also believed that if a person was unjust, inhumane, or disrespectful, he would not be protected but would instead be punished by the gods.

Katō further argues that over history, there has been a belief that gods protect the forces with justice. Still, there has also been an added belief that those who die in battle following the forces with justice can become gods of national protection. He cites the existence of Yasukuni Shrine as the "most realistic example" of this belief and outlines its history (Katō 1938, p. 61).

*4.3. "The Buddhist View of War"*

In Section 3, "The Buddhist View of War", Katō's first question was about the existence of "demons". This is also related to the issue of justice raised in the previous chapter. Here, Katō started his discussion with the process of the emergence of religion in general before starting the discussion of Buddhist doctrines. Applying comparative religious findings such as references to primitive beliefs, Judeo-Christian tradition, Zoroastrianism, and Greek and Roman mythologies, he points out that war conducted by gods developed to overcome evil, represented as a demon.

He goes on to explain that Buddhism is also a religion of conquering demons but that its main target is not the demons that roam the outer world but the afflictions as demons that leap within the mind and that the process of perfecting one's personality through cultivation is defeating demons (Katō 1938, pp. 70–71).

However, in the latter half of this chapter, the question is how Buddhism views real wars, which are not "wars within the mind" but wars involving human beings killing each other. First, Katō stated that religions generally preach happiness and that peace is happier than war. The history of humanity has been filled with conflicts, and the history of religions that should be peaceful often has a bloody past. He asked whether Buddhism endorses war even if there are fewer wars in the history of Buddhism.

Drawing on various Buddhist scriptures, he explains that the highest ideal status is to secure peace through non-resistance. However, in reaching this stage, it is inevitable to engage in battles, and Buddhism accepts raising arms and fighting to protect the country and nurture the people. In other words, the Buddhist view of war is that it is permissible to wage war based on righteousness and sound guidance.

Katō also raised the issue of the relationship between war and the precept of non-killing, citing killing as the most severe Buddhist precept, along with theft, lewdness, and falsehood. He introduced the Buddhist scriptures, which state that "to kill an evil person with a good heart is less sinful than to kill an ant with an evil heart" and that "to kill a person who is harmful to the nation is without sin" (Katō 1938, p. 88).

In the divine land thought of Section 2 and the Buddhist theory of Section 3, the respective religious worldviews and doctrines lead to the understanding that the war with justice is acceptable. These understandings can be read as corresponding to his statement at the end of Section 1 regarding the ongoing Sino-Japanese War, "We sought peace, they sought war, and at last, the injustice that provoked this incident was theirs, and it is needless

to say that the fault also lies with them" (Katō 1938, p. 8). Thus, the religious-historical defense of the Japanese military's conduct of the war is the focus of the discussion.

The original text for this chapter can be found in one of Kato's earlier works, *Gōmahyō kōwa: Kokoro no tatakai* 降魔表講話—心の闘い (*The Lecture on the Report of defeating demons: the battle within the mind*) (Katō 1935). The book is a record of the lecture given by Katō to military personnel on 19 October 1933 and was first published in the monthly report of the lecture series, which was available only to army generals (Katō 1933; Tzeng 2019). Kato's speech activities during this period were directed both to the military personnel who were going into battle and to the general public who were defending in the all-out war system.

*4.4. "Samurai and Faith" and "Emerging Spirit and Faith"*

The next chapter, "Samurai and Faith", focuses on the samurai's rise as fighters in Japanese history from ancient times to the Middle Ages and the specific aspects of their faith. The author explained the influence of the samurai on Japanese religion by introducing the moralistic attitude of the warriors. They were devoutly religious and respected the gods and Buddhas, and their martyrdom for their faith behind the religious use of force in the Ikkō Rebellion 一向一揆 and the Hokke Rebellion 法華一揆 with Buddhist faiths and the Amakusa Rebellion 天草一揆, led by the Christians. It also discusses how religious inspiration encouraged the warriors to fight at the risk of their lives with an unwavering spirit.

Section 5, "Emerging Spirit and Faith", focuses on Japan's conduct of the war in the modern era. In response to the ridicule of Japan as a belligerent nation, Katō mentions that at many periods in history, there were no wars as seen in other countries. He argues that Japan "was never a warlike people" but a "people who enjoyed peace" (Katō 1938, p. 159). He emphasizes that they are not a nation that raises an army in vain, saying that they fight when they have no choice but to fight and are prepared to respond to attacks. He then argues that during the Edo period, a national awareness arose that prepared the nation for the transition from a feudalistic ideology to a modern emperor state. Furthermore, after the Meiji period, because of the dismantling of the samurai class and the formation of a modern military under the universal conscription, the mind of national unity and the wartime awareness of the people through wartime mobilization became widespread.

As a social trend that causes cracks in this mind of national unity, he cited the influence of modern thought that accompanied the West's expansion into the East and urged caution. Specifically, he referred to materialistic commercialism with indifference to religion and spirit, resulting in selfish individualism and the revolutionary ideas of equality. Katō saw these as undermining the unity of the nation and society by prioritizing individual and class interests. In addition, he stated that the international spread of class struggle and the emergence of communism, which aimed to transform the social organization, from an anti-religious perspective, and viewed humanity only in material terms and that China at the time was at the mercy of this view, as it inhibited spiritual well-being. Katō argued that the Buddhist doctrine of the correlation of all things in the universe and the doctrine of nondiscrimination (equality of all living things) explains the totalitarianism promoted to counter Western thought. He stated that "the Buddhist doctrine of the reality of the universe is an iron law that we must not forget in our national life as well" (Katō 1938, p. 173).

Katō saw modern warfare as a competition between national powers, not between weapons, and stressed the importance of a system of total spiritual mobilization that unites the people of Japan after the war and the soldiers who went to war. He also emphasized that the central idea of these virtues is sincerity and that firm faith is necessary.

Here, he refers to the discussion of the emergence of religious faith in religious studies as "religious faith is so powerful that it can indeed transform human life" (Katō 1938, p. 182). However, it arises from the fact that humans have a latent religious sentiment that yearns for the absolute existence of God and Buddha.

As seen in this section, the theme of Sections 4 and 5 of Katō's *War and Faith* can be understood as an assertion that faith plays a significant role in establishing a steadfast spirit in the face of war. The book's emphasis on the "fearful rearguard disturbance caused by idle rumors" and its call for enduring difficulties with "steadfast faith unmoved by them" (Katō 1938, p. 184) indicate that the intended audience of the book was not combatants but the public who would support the total war system in the rear.

## 5. Confronting Death

Six years after the discussion in *War and Faith*, published in 1938, *Confronting death* was published in February 1944, when the Japanese military was on the backfoot in Asia and the Pacific and air raids on the Japanese mainland were already imminent (Katō 1944). As the title suggests, this book is about death. As noted in the "Words of the Author," Katō had already published *Shisei mondai* 死生問題 (*The issue of death and life)* and *Shigo wa dounaru* 死後はどうなる (*What will happen after* death) (Katō 1932, 1936), and there is much overlap in East–West comparisons and quotations from poetry. He also published three books on "views of life and death" in the 1900s (Katō 1904, 1905, 1908), as Shimazono mentions in Section 2 of this paper; however, the general framework of an East–West comparison of changes in views of life and death was followed. In *Confronting death*, Katō focuses on "immediate preparedness rather than postmortem issues" (Katō 1944, p. 2), suggesting that this work was extremely conscious of the wartime situation.

This becomes clearer when we review the intent of the publication of this book. In the "Author's Words," Katō states that he "seeks a model for addressing emergencies and dealing with a decisive war" (Katō 1944, p. 1). In the "Concluding Remarks," he refers to his desire not to fail to use the footsteps of his predecessors in the training of "kokumin shikon 国民士魂[5]" (Katō 1944, p. 227). This new concept, which appears at the end of the book, is not explained in that text[6], but in the previous year, 1943, he published a book under the title *Kokumin shikon* (Katō 1943). At the beginning of the latter book, he wrote, "Under the system of total national war, every citizen must be prepared to serve as a soldier" (Katō 1943, p. 1). He describes the ethical character of the broadness of mind and strength of will as the true nature of a warrior and an unchanging, strong, and pure state of mind as the basis of the spirit (Katō 1943, pp. 3, 9).

How, then, is the issue of preparedness for death discussed? The author cites exemplary figures who faced death with determination, known as "Shishi 志士" or "Jinjin 仁人", and argues that they did not choose death because they renounced life or longed for a life after death. Instead, it is because there were things that they valued more than their individual lives or something they were willing to do to society more than avoiding death. He tried to approach the central theme of readiness for death through the achievements of people who were martyrs to the nation, religion, morality, honor, freedom, truth, and other noble ideals and through the poetry they left behind on a level that transcended the issues of life and death.

*Confronting death* is organized as follows: Section 1, "Introduction", followed by Section 2, "Talking about the West", and Section 3, "Talking about China". Sections 4 and 5, "Talking about Japan (1 and 2)", in which the main content compares the historical development of the view of life and death in the East and West. Regarding the flow of the discussion from the West to the East and then to Japan, Katō states, "It is nothing less than an attempt to clarify the characteristics of the Japanese spirit, extending from the remote to the near" (Katō 1944, p. 6). Regarding Western examples, he discusses a wide range of topics from ancient Greece and Rome, chivalry and crusades in medieval feudal society, the Reformation, the scientific elucidation of truth by Copernicus and others, and slogans associated with modern ideals, etc. He also discusses many historical events and ideological episodes in China. Still, the comparison is not intended to relate to Japan's view of life and death but rather to exalt Japan's superiority. In the "Words of the Author" at the beginning of the book, the author elucidates that in the West, deaths of loyalty are rare. In

China, martyrdom is occasional, with only several examples found in each region, while such an attitude is uniformly found only in Japan.

The flow of content, which begins in the West, moves through East Asia and converges with descriptions of Japan and historical changes from ancient times to the same period, reaching its climax concerning the audacious attack 玉砕 discussed at the end of the book. The book emphasized that even if the "flesh" of each demise, they will live on as a loyal soul who supports the *Kokutai* 国体 (the national entity) in a manner unmatched by any other nation. The author concludes that the Meiji Restoration was achieved through such mettle. The term Japanese bushidō 武士道 became known throughout the world in modern foreign wars, and in the Pacific War, "the assault on Pearl Harbor", "deaths in the South Pacific", and "the crushing of the Japanese flag on the island of Attu" (Katō 1944, p. 227).

## 6. Conclusions

As mentioned in the introduction of this paper, David Chidester pointed to the application of 19th century scholarship on comparative religion to South Africa's apartheid policies in *Savage Systems* (Chidester 1996). On the other hand, the East–West comparison of views on life and death in *Confronting death* discussed in the second half of this paper can be characterized as a wartime appropriation of comparative religion. The method of comparison, in this case, does not use religious or cultural differences as indicators of racial segregation, as in the case of apartheid. Instead, it emphasized Japan's superiority over other countries by highlighting the continuity of the imperial regime from the mythological period and the spirit of martyrdom. Although *War and Faith* did not employ the East–West comparative approach in its chapter structure as in *Confronting death*, the same effect was expected when applying comparative religion's findings and describing Japanese religious characteristics based on the assumption of religion in general.

What, then, was the expected effect? It is clear from the language and distribution of the two books that they were intended for a wartime audience of Japanese readers. They were designed to agitate Japanese readers and encourage them to participate proactively in the fight to the death by mobilizing their spirit under the total war system. Therefore, Katō's wartime discourse discussed in this paper can be described as the agency of comparative religion in total war.

In Japan, the pros and cons of military research are currently being actively debated at the Science Council of Japan and various universities (Sensō shakaigaku kenkyūkai 2020). Rather than forgetting the history of how religious analysis was once used for military research, it is necessary to reassess the issue critically. This paper is one of the essential works for this purpose.

**Funding:** This research received no external funding.

**Institutional Review Board Statement:** Not applicable.

**Informed Consent Statement:** Not applicable.

**Data Availability Statement:** The document of (Katō 1933) is available at JACAR (Japan Center for Asian Historical Records, https://www.jacar.go.jp/, accessed on 25 May 2022) Ref. C15120199100. The other Katō's texts are available at National Diet Library Digital Collections (https://dl.ndl.go.jp/, accessed on 25 May 2022).

**Conflicts of Interest:** The author declares no conflict of interest.

## Notes

[1] For example, Katō published *Kokuminsei to Shūkyō: Tsūzoku kōwa* 国民性と宗教—通俗講話 (*Nationality and Religion: Public Lecture*) (Katō 1915). At the beginning of the book, Katō argues that in order to study nationality 国民性 (or national thought 国民思想), it is necessary to observe their religions that has cultivated its basis. Since the book is intended for the indoctrination of the public, he states that he has tried to avoid lofty and to describe it in a simple manner, omitting abstrusity. At the middle section of the book, he introduced the study of religion including both the history of religion and comparative religion. Katō introduces academic classifications of the world's religions, such as polytheism, monotheism, and pantheism, with their English terms. On

the other hand, as his own view, he explains three religious currents, Semitic, Aryan, and Chinese, and he argues that Japan is an exhibition hall of world religions, where all three currents have been introduced. Furthermore, he explained the contemporary Japanese religions, such as Shintō sects, including the new religions like Konkō-kyō 金光教 and Tenri-kyō 天理教, which have emerged since the late Edo period, as well as Buddhist sects and Christian churches. He referred to the study of folk beliefs begun by Anesaki (Anesaki 1897). Katō even wrote another book on the history of folk beliefs (Katō 1925), which also shows he took a role of an introducer of religious studies to the public.

[2]   The subject of the influence of religious studies on war, which is examined in this paper, is necessarily related to the larger question of how the ideas and practices of religions were involved in the war. Although the relationship between the two issues must be examined in further research, noteworthy works have been recently published on the subject of religion and war. See (Ogawara 2010, 2014; Ishikawa 2013; Nagaoka 2015; Ōsawa 2015; Sensō shakaigaku kenkyūkai 2019; Hirafuji 2020; Shimazono et al. 2021; Rekishikagakukyōgikai 2021).

[3]   The abbot of the Honganji subsect of Jōdo Shinshū Buddhism, an explorer.

[4]   E.g., the books by Ōkawa Shūmei 大川周明 (1886–1959), Kanokogi Kazunobu 鹿子木員信 (1884–1949), and Kōno Seizō 河野省三 (1882–1963), Imaizumi Sadasuke 今泉定助 (1863–1944), et al.

[5]   It is difficult to translate this concept. *Kokumin* 国民 means nation and *Shikon* 士魂 the spirit of the samurai. Therefore, it implies an entire national population imbued with the spirit of the samurai.

[6]   In the text, references to the expressions "shifū 士風 (samurai ethos)" and "shidou 士道 (the way of virtue as a samurai, bushido)" in each period are acknowledged.

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
