# Peer review of "Applying Religious Studies Discourse during Wartime: On Katō Totsudō’s Discussion of Religious War"

_religions, doi:10.3390/rel13060533_

Round 1

Reviewer 1 Report

This is an interesting discussion that could be made more compelling by bringing some of the issues raised in the conclusion to the introduction, so as to provide a wider context for the reader. Specifically, the comparison with Chidester's work deserves some expansion at the beginning along with much more information about current discussion of these topics in Japan. Nothing from the extensive recent studies of Buddhism and war/violence are brought to bear, so that the essay remains simply expository. The first recommendation here—providing better context by referring to issues now only found in the conclusion in the Introduction as well—is a minor revision, which is what I will recommend; integrating more contemporary scholarship on Buddhism/religion and war/violence would make it of much more use to many more scholars, but would be more of a major revision. The latter is desireable, but not necessary.

Author Response

Thank you so much for your very thought-provoking comments. Considering this, I tried to present the reader with a broader scholarly context by clearly stating in the introduction the issues referred to in the concluding section. While your last part comments are undoubtedly correct, I could not digest the recent research on the involvement of religious thought and sects in the war due to my focus on the impact of the scholarly discourse on the war. Instead, I have only included several notes on important references in recent studies.

Reviewer 2 Report

Apart from some minor problems with the English, this is an excellent essay that makes an important and original contribution to the study of the propagandistic use of religion in wartime.

Author Response

I appreciate your positive feedback. Based on suggestions from other reviewers, I made the following main revisions.

  1. I presented a broader scholarly context for the reader by headlining the concluding discussion points in the introduction.
  2. A brief introduction to the latest research trends in Buddhism/religion and war/violence is included in the notes.
  3. The introduction of Kato's detailed bio and the influence of religious studies discourses on the discourse.
  4. The use of consistent notation and more appropriate translations

Reviewer 3 Report

General Comments 

This paper engages in a detailed case study to consider an important question: How the academic discipline of religious studies and its (popularized) discourse was enlisted to arouse popular support for the war effort during the 1931-1945 conflict. During this period virtually every system of thought or belief, from various academic disciplines through organized and organic belief systems, was compelled to find some form of accommodation with the official wartime ideology. Many responded to this challenge with enthusiasm; others with greater reluctance. Those that failed entirely to integrate the dominant ideology were often silenced or suppressed. By focusing on how a popular and influential  lay Buddhist and writer aligned various concepts and ideas from his tradition with the war aims, this paper contributes to the existing literature on the role of Japanese Buddhism (and more largely “religion”) within the prevailing ethos of wartime. It adds important and interesting details through a close reading of several of Katō’s texts. 

In terms of the stated objective of demonstrating how the religious studies discourse was “applied” and “understanding the influence of religious theory and religious studies discourse on Japanese society during the war,” however, the results are less clear. While the presence of elements of religious studies, such as comparative religion, is demonstrated, its influence is not examined in concrete detail. The author is frank about the reasons for this: “Of course, it is also necessary to examine reactions  from the recipient side, but due to material limitations, this point will not be addressed in  this paper.” This limitation is specifically noted with regard to the military schools where it is thought that Katō’s texts may have been referenced, but in the case of Buddhist View of War (“Emerging Spirit and Faith”), the author notes the reference to the “fearful rearguard disturbance caused  by idle rumors,” thus surmising that the intended readership was the public rather than combatants. Here the fact that Katō’s works generally sold well and were widely read suggests their scope of impact and influence. This is important and, as noted above, helps fill in more details of the wartime semantic space in which ideological content was generated and delivered to different audiences. 

Since analysis from the recipient side has been put off for later, any analysis of the application or influence of religious studies discourse in Katō’s work must rely on the evidence in the texts themselves. Here, unfortunately, what is presented is rather scant. Several examples of a comparative religions approach are given, as are references to a global (putatively universal) history and taxonomy of religion, but these do not appear to play a key role in Katō’s arguments. Rather, they seem to serve at most as a framing backdrop for the  main task of convincing either soldiers or civilians of the legitimacy of self-sacrifice in the name of the state, argumentation that centers on Katō’s reading of Buddhist texts or some form of Japanese cultural essentialism. If there are instances of more specific and polemically central citations of religious studies discourse in Katō’s texts, their inclusion would strengthen the paper’s pursuit of its stated objectives. 

Other Comments

Biographical sketch 

The text as it stands assumes a considerable prior knowledge of Katō’s life and work, which may be true for some readers, but not others. While biographical information is presented throughout the paper, a brief initial sketch, including more details (if available) of his interactions with Anesaki and the emerging field of religious studies, would help readers orient themselves and follow the author’s arguments. 

Name order, macrons, and romanization

Name order is inconsistent, as the abstract gives Katō’s names in both Japanese and Western order: Totsudō Katō and Katō Yū’ichirō. Placing all names in Japanese order (family, given) would probably be best for the journal’s readership. 

Likewise, a review of macrons to ensure they are always used for long vowels in Japanese is recommended (such as for Showa and Taisho). Likewise, successive vowels in Japanese names and terms are separated by commas to indicate that they are pronounced individually and not as diphthongs. In some cases, such as Yū’ichirō this is helpful, but in others, such as Ra’i’ō Nishizawa, Ma’ekawa, and Shise’ikan, this seems to be nonstandard and distracting. The journal’s copy editing staff may be able to provide guidance in this regard. 

Translated terms 

Certain terms are translated in ways that don’t convey their full weight. 国体(at the time: 國體) is translated as “polity,” which is adequate in many contexts, but in the period in question this term had an ideological centrality and talismanic power that is hard to overstate. This is sometimes translated as “national essence” or “national entity,” etc. and whichever translation is used, some explanation of the background of this term and its implications of absolute loyalty to the emperor, would be helpful. “Rebellion” or “uprising” might be better renderings of 一揆 than Putsch. Likewise “nation master spirit” doesn’t seem to convey the full meaning of 国民士魂. Here also some explanation of the characters used and seeming implication of an entire national population imbued with the spirit of the samurai, would be helpful to readers.

Author Response

Thank you so much for your detailed comments. In terms of the impact on readers, you pointed out the issue of the effect on the public. The influence of religious theories and discourses is indeed limited, which appears in the war-time texts themselves. I could not develop a sufficiently convincing explanation of your point of view. Therefore, I made the following modifications.

1) By pointing out the existence of Kato's lectures to military personnel, which may have been the source of the description of the "Buddhist view of war" in Chapter 3 of War and Faith discussed in Section 4.3. By suggesting the existence of a specific audience within the military, not just the general readers of the book, I sought to explain the context more clearly to the readers of this paper.

2) Regarding your latter point about the importance of religious history and religious theories for Katō, I could not find any passages that make this clear in Kato's texts during the war. In addition, he did not intend for specialized intellectuals but the public as readers, and therefore, highly abstract discussions of religious theories do not appear in them. Instead, I referred to the earlier text "Nationality and Religion" as an excellent example of his explanation of religious theories for the public. By introducing it and more carefully submitting Kato's discussion of folk beliefs mentioned therein, I will be able to supplement the readers of this paper with an explanation of his relationship to Anesaki Masaharu and the relationship between religious studies discourse and propaganda discourse to the public.

In terms of the other comments, I revised them as follows.

Biographical sketch: I have added a supplemental explanation to the introductory section based on the discussion of previous studies.

Name order, macrons, and romanization: Thank you for your remarks about the order of names, macrons, and romanization. In the revision, I have made the corrections of them.

Translated terms: All you pointed out were persuasive, and I have revised the draft in your direction.